

# Comparative transcriptome analysis of panicle development under heat stress in two rice (*Oryza sativa* L.) cultivars differing in heat tolerance

Yaliang Wang[*], Yikai Zhang[*], Qiang Zhang, Yongtao Cui, Jing Xiang, Huizhe Chen, Guohui Hu, Yanhua Chen, Xiaodan Wang, Defeng Zhu and Yuping Zhang

State Key Laboratory of Rice Biology, China National Rice Research Inistitute, Hangzhou, Zhejiang, China
[*] These authors contributed equally to this work.

## ABSTRACT

Heat stress inhibits rice panicle development and reduces the spikelet number per panicle. This study investigated the mechanism involved in heat-induced damage to panicle development and spikelet formation in rice cultivars that differ in heat tolerance. Transcriptome data from developing panicles grown at 40 °C or 32 °C were compared for two rice cultivars: heat-tolerant Huanghuazhan and heat-susceptible IR36. Of the differentially expressed genes (DEGs), 4,070 heat stress-responsive genes were identified, including 1,688 heat-resistant-cultivar-related genes (RHR), 707 heat-susceptible-cultivar-related genes (SHR), and 1,675 common heat stress-responsive genes (CHR). A Gene Ontology (GO) analysis showed that the DEGs in the RHR category were significantly enriched in 54 gene ontology terms, some of which improved heat tolerance, including those in the WRKY, HD-ZIP, ERF, and MADS transcription factor families. A Kyoto Encyclopedia of Genes and Genomes (KEGG) analysis showed that the DEGs in the RHR and SHR categories were enriched in 15 and 11 significant metabolic pathways, respectively. Improved signal transduction capabilities of endogenous hormones under high temperature seemed to promote heat tolerance, while impaired starch and sucrose metabolism under high temperature might have inhibited young panicle development. Our transcriptome analysis provides insights into the different molecular mechanisms of heat stress tolerance in developing rice.

## INTRODUCTION

Climate change is predicted to increase the average global temperatures by 0.3–4.8 °C by the end of the 21st century (*Stocher et al., 2013*). Unusually high temperatures occur frequently during the rice growing season (*Dwivedi et al., 2015*; *Tao et al., 2013*), and cause reductions in the yield and quality in several rice producing regions, including China, India, and Japan (*Anand, Kumar & Narayan, 2018*; *Morita, Wada & Matsue, 2016*; *Wang et al., 2019*). The primary cause of rice yield reductions is a reduction in spikelet fertility due to high temperatures during the flowering period (*Espe et al., 2017*). Rice quality is also influenced

Corresponding authors
Defeng Zhu, cnrice@qq.com
Yuping Zhang, cnrrizyp@163.com

by high temperature, which causes carbohydrate metabolism disorders (*Yamakawa & Hakata, 2010*). As climate change has intensified, extremely high temperatures above 40 °C have become more frequent. Such high temperatures inhibit rice panicle development, reduce the spikelet number by 5%–15%, and aggravate rice yield losses (*Wang et al., 2017*).

High temperatures adversely affect floral development by reducing the antioxidant capacity, inhibiting nutrient accumulation, and degenerating tapetal cells (*Prasad, Bheemanahalli & Jagadish, 2017*). A previous study showed that high temperature (39 °C) conditions downregulated certain genes related to tapetum function, pollen adhesion, and germination, including *OsINV4* and *OsMST8*, which influenced spikelet fertilization (*Endo et al., 2009*). In addition, sugar and endogenous hormone metabolism under high temperature reportedly plays an important role in pollen formation in both rice and cotton (*Islam et al., 2018*; *Min et al., 2014*). At the rice ripening stage, high temperature induces early termination of grain filling (*Kim et al., 2011*). Grain chalkiness increases under a mean temperature greater than 32 °C, resulting in the deterioration of eating and cooking quality, which are both closely linked to starch and sucrose metabolism (*Zhong et al., 2010*). Transcriptome analysis has shown that high temperatures influence the expression of genes involved in the inhibition of sucrose degradation and starch biosynthesis while promoting starch degradation and storage proteins synthesis (*Yamakawa & Hakata, 2010*; *Yamakawa et al., 2007*). *Takehara et al. (2018)* reported that the upregulation of *OsSUS3*, which encodes sucrose synthase, improved high-temperature tolerance.

The panicle initiation stage is an important period of spikelet proliferation. Dry matter accumulation is essential for panicle development; however, the pathway for carbohydrate accumulation during spikelet formation under heat stress remains vague. The reduction in spikelet number that occurs under high temperature conditions has been associated with heat-induced phytohormone changes, especially enhanced cytokinin degradation (*Wu et al., 2017*; *Wu et al., 2016*). The number of spikelets per panicle is determined by spikelet differentiation and degeneration. Spikelet differentiation is correlated with dry matter accumulation and influenced by environmental factors (*Liu et al., 2005*). *Ding, Wang & Ding (2016)* reported that hormone metabolism, the stress response, carbohydrate metabolism and transport, and protein degradation were regulated to influence panicle initiation. Additionally, certain genes, such as MADS-box genes, are related to panicle initiation (*Kang et al., 2013*; *Kobayashi et al., 2012*). Quantitative trait loci for spikelet degeneration have been identified (*Yamagishi et al., 2004*), and the genes *SP1*, *ASP1*, *TUT1*, *PAA2*, and *OsALMT7* have been found to control spikelet degeneration (*Bai et al., 2015*; *Heng et al., 2018*; *Li et al., 2010*). However, the mechanism of panicle development under high temperature conditions is still unclear. In this study, an RNA-Seq analysis was used to explore the mechanism of heat tolerance during panicle development. Huanghuazhan (HHZ) is a heat-tolerant rice cultivar widely grown in the middle and lower reaches of the Yangtze River in China (*Cao et al., 2009*; *Zhou et al., 2012*). IR36 is a heat-susceptible cultivar (*Fang, Tang & Wang, 2006*) and a parental line of HHZ. In the current study, we investigated transcriptome differences between these two cultivars exposed to different temperatures 40 °C and 32 °C, during the spikelet differentiation stage. We identified differentially expressed genes (DEGs) in young panicles of the two cultivars under the

two temperature treatments and performed Gene Ontology (GO) enrichment and the Kyoto Encyclopedia of Genes and Genomes (KEGG) analysis. This work improves our understanding of the molecular mechanism underlying the heat-induced inhibition of spikelet development and provides important insights into rice breeding.

## MATERIAL AND METHODS

### Plant materials and heat stress treatments

We used the rice cultivars HHZ and IR36 in this study. Pre-germinated seeds were sown in seed trays filled with a mixture of vermiculite (20%), charcoal (30%), soil (40%), and slow-release fertilizer (10%). After 20 days, the seedlings were transplanted into pots with four seedlings per pot. Each pot (24 cm length × 22.5 cm width × 21.5 cm height) contained 10 kg air-dried paddy soil. Pots were kept under natural environmental conditions (the average temperature was 30–35 °C).

Before seedlings were transplanted into pots, fertilizer was applied to each pot based on a field application rate of 14 kg nitrogen per 666.7 $m^2$. Before transplanting into the pots, 3.5 g compound fertilizer (nitrogen: phosphorus: potassium = 15%: 15%: 15%) was applied to each pot. At the tillering stage, 0.6 g urea was supplemented in each pot. At panicle initiation, 0.6 g urea and 0.5 g potassium chloride were also applied to each pot. Pests, diseases, and weeds were intensively controlled.

Automatic growth chambers (Qiushi Environment Corporation, Hangzhou, China) were used to conduct the temperature treatments. Plants were moved to the growth chambers on the approximate date of spikelet differentiation when the panicle length was approximately 0.2 cm (around 60–70 d after seed sowing). The high-temperature (40 °C) and control temperature (32 °C) treatments were implemented for eight hours each day from 9:30 to 17:30 h (the setting details are shown in Table S1) for nine days. The humidity in the chambers was maintained at 75–80%. Rice plants were grown under natural ambient conditions during all growth stages before and after the temperature treatments. Each treatment contained three replicates with 20 pots per replicate.

### Panicle and spikelet morphology

Ten main tillers were sampled per replicate on day 9 of treatment at 40 °C or 32 °C to investigate the development of young panicles at high temperature.

Spikelet differentiation or degeneration of the main tiller panicles was determined at the heading stage. The number of degenerated spikelets was calculated by counting the vestiges present on the panicles. The number of differentiated spikelets was the sum of the surviving and degenerated spikelets. The proportion of degenerated spikelets was then calculated.

Spikelet morphology was observed under a stereomicroscope (Olympus SZX7, Olympus Corporation, Tokyo, Japan) and the glume length and width (mm) were measured at 0.63x and 2.5x using the microscale in the Image Pro-Plus 5.1 image processing software (Olympus SZX7; Olympus Corporation, Tokyo, Japan). Fifteen spikelets were collected from the upper, middle and lower parts of each panicle, with five panicles sampled for each replicate.

## RNA extraction, transcriptome sequencing, and mapping

After nine days at the 40 °C or 32 °C treatment, young panicles from 20 main tillers were collected for each replicate at 12:00–13:00 and immediately frozen in liquid nitrogen. In quick succession, TRIzol reagent (Invitrogen, Carlsbad, CA, USA) was used to extract total RNA from the young panicles according to the manufacturer's instructions. A TruSeq RNA Sample Preparation Kit (Illumina Inc., San Diego, CA, USA) was used to generate 12 sequencing libraries according to the manufacturer's instructions. The sequencing libraries were then sequenced on a HiSeq platform (Illumina, Inc., CA, USA). High-quality sequence reads were obtained by filtering the raw data and then compared to the 9311-reference genome (Oryza_indica.ASM465v1.dna.toplevel. fa) obtained from http://www.ensembl.org/. The raw RNA sequence data were submitted to the NCBI Sequence Read Archive with accession number PRJNA508820.

## Gene expression level and differential expression analysis

We used HTSeq (0.9.1) to statistically compare the read count values of each gene, which represent the original expression of each gene. Fragments per kilobase of transcript per million mapped reads (FPKM) was used to standardize the expression. Next, we used DESeq (1.30.0) to analyze the differential expression of genes with the following screening conditions: an expression difference of |log2foldChange | > 1 and a significant $P$-value < 0.05.

## GO and KEGG enrichment analysis of DEGs

For the GO enrichment analysis of DEGs, we used the Singular Enrichment Analysis tool in AgriGO (http://bioinfo.cau.edu.cn/agriGO/analysis.php) with the default parameters, and a $P$-value ≤0.05 indicated significant enrichment. The KEGG enrichment analysis of DEGs was performed using KOBAS software with default parameters and a $P$-value ≤ 0.05 indicated significant pathway enrichment.

## Verification of RNA-Seq by quantitative real-time PCR (qRT-PCR)

First-strand cDNA was synthesized using ReverTra Ace qPCR RT Master Mix with gDNA Remover (Toyobo, Osaka, Japan) according to the manufacturer's instructions. The qRT-PCR analyses were performed using an Applied Biosystems 7500 Real-Time PCR system with Power SYBR Green PCR Master Mix (Applied Biosystems, Carlsbad, CA, USA). The primers used for qRT-PCR are listed in Table S2. The *OsUBQ* gene was used as an internal control. Relative gene expression levels were determined from the equation $2^{-\Delta\Delta CT}$ (*Czechowski et al., 2004*), where $\Delta\Delta CT$ represents $\Delta CT$ (target gene of interest)–$\Delta CT$ (control gene).

## Statistical analyses

Microsoft Excel 2016 (Microsoft Inc., Redmond, WA, USA) was employed for data collection. The panicle and spikelet morphological data collected for the 40 °C and 32 °C treatments (mean of three replicates) were statistically analyzed by Student's $t$-test ($P < 0.05$). Graphs were created using Origin 9.1 (Ver. 9.1; OriginLab, Northampton, MA, USA).

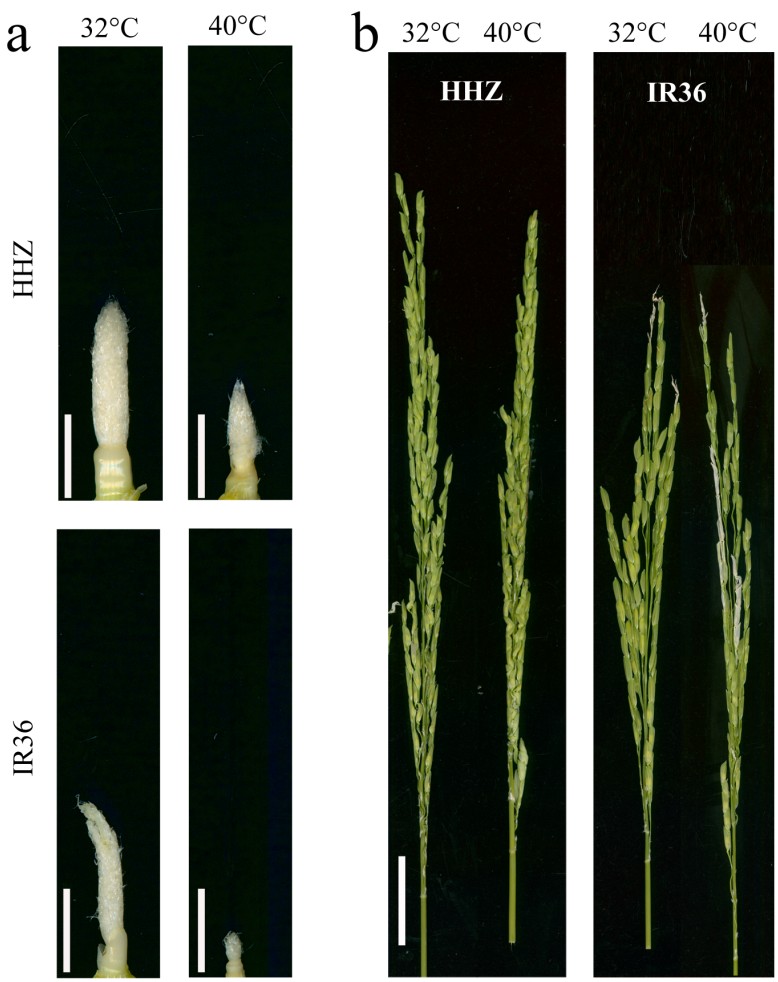

**Figure 1** **Effects of high temperature on panicle development.** (A) Young panicle morphologies after 9 d of high-temperature treatment; and (B) panicle morphologies at the heading stage after high-temperature treatment. Bars = 0.5 cm in (A) and 3 cm in (B).

## RESULTS

### Spikelet development at high temperature

A preliminary experiment showed a significant difference in panicle development, which was measured as spikelet differentiation, after nine days of the high-temperature treatment. The results reported in the current study are consistent with these preliminary findings. Spikelet differentiation inhibited young panicle growth after nine days of the high temperature treatment (Fig. 1). After the temperature treatments, the panicles required an additional 15–20 days to complete growth.

Compared to the control temperature treatment, the high temperature treatment reduced spikelet survival by 22.3% ($P < 0.05$) for HHZ and 53.6% ($P < 0.05$) for IR36. With high temperature, the number of differentiated spikelets decreased by 9.6% and 33.2% ($P < 0.05$) for HHZ and IR36, respectively, and the proportion of degenerated
spikelets significantly increased by 32.3% ($P < 0.05$) and 67.4% ($P < 0.05$), respectively. In addition, the heat treatment reduced the glume length by 10.3% ($P < 0.05$) for HHZ and by 16.0% ($P < 0.05$) for IR36 and reduced the glume width by 12.0% ($P < 0.0.5$) and 8.0% ($P < 0.05$), respectively. The reductions in spikelet number and size led to reductions in panicle weight of 33.2% ($P < 0.05$) for HHZ and 67.7% ($P < 0.05$) for IR36. The larger reduction in panicle weight in IR36 suggests that high temperature has a greater effect on young panicle development in heat susceptible cultivars (Table 1).

## Transcriptome analysis

Under the 32 °C control temperature, a total of 44.2 million and 48.9 million raw reads were obtained from HHZ (referred to as HHZ_32) and IR36 (referred to as IR36_32), respectively. Under the 40 °C treatment, a total of 45.5 million raw reads were obtained from both HHZ (HHZ_40) and IR36 (IR36_40) (Table 2 and Table S3). More than 99.0% clean reads were obtained for the downstream analysis. The results of RNA sequence mapping indicated that 85.8–88.0% of the clean reads could be mapped onto the reference genome and most were uniquely mapped (Table 2).

## Identification of DEGs

To compare the differences between the two cultivars at 40 °C and 32 °C, we used four comparison groups: HHZ_32 vs. HHZ_40, IR36_32 vs. IR36_40, IR36_40 vs. HHZ_40, and IR36_32 vs. HHZ_32. DEGs for the four groups were restricted to those with a |log2fold change|>1 and a *P-value* < 0.05. With these criteria, 3,342, 2,469, 2,949, and 2,461 DEGs were detected for HHZ_32 vs. HHZ_40, IR36_32 vs. IR36_40, IR36_40 vs. HHZ_40, and IR36_32 vs. HHZ_32, respectively (Fig. 2). Significantly different gene expression was observed both between cultivars and between treatments. For HHZ, 1,794 genes were upregulated and 1,548 genes were downregulated in the 40 °C treatment compared with the 32 °C treatment (Fig. 2). Furthermore, 1,140 genes were upregulated and 1,329 genes were downregulated in IR36 under the 40 °C treatment compared with the 32 °C treatment (Fig. 2). For comparisons within treatments, 1,408 genes were upregulated and 1,541 were downregulated in the IR36_40 vs. HHZ_40 group and 893 genes were upregulated and 1,751 genes were downregulated in the IR36_32 vs. HHZ_32 group (Figs. 2C and 2D).

## Classification of DEGs

In all four groups, a total of 5,533 unique DEGs were identified, and they could be divided into 15 disjointed subgroups (Fig. 3). Among the 15 subgroups, eight from the IR36_32 vs. HHZ_32 group were excluded from the analysis because they were not influenced by high temperature. In addition, 1,157, 603, 524, and 402 DEGs were uniquely identified in the HHZ_32 vs. HHZ_40, IR36_32 vs. IR36_40, IR36_40 vs. HHZ_40, and IR36_32 vs. HHZ_32 groups, respectively. The DEGs in groups that were responsive to high temperature could be further sorted into three categories: heat-tolerance-cultivar-related genes (RHR, 1,688 genes), heat-susceptible-cultivar-related genes (SHR, 707 genes), and common heat stress-response genes (CHR, 1,675 genes) (Table 3 and Table S4). The DEGs in the RHR category might have played an important role in heat tolerance, whereas the DEGs in the SHR category might be associated with heat injuries in the heat-susceptible cultivar.

Wang et al. (2019), *PeerJ*, DOI 10.7717/peerj.7595

**Table 1 Panicle characters after high temperature treatment.**

| Cultivars | Treatment | Panicle weight(g) | Spikelet number | The number of differentiated spikelet | The proportion of degenerated spikelet (%) | Spikelet fertility (%) | Grain weight (mg) | Glume length (mm) | Glume width (mm) |
|---|---|---|---|---|---|---|---|---|---|
| HHZ | 32 °C | 3.6 ± 0.4 | 235.0 ± 20.0 | 335.3 ± 20.5 | 30.0 ± 1.7 | 83.0 ± 2.0 | 18.5 ± 0.3 | 8.7 ± 0.2 | 2.5 ± 0.0 |
|  | 40 °C | 2.4 ± 0.3[**] | 182.7 ± 11.2[**] | 303.3 ± 10.1 | 39.7 ± 5.3 | 78.5 ± 1.3[**] | 17.0 ± 0.3[**] | 7.8 ± 0.1[**] | 2.2 ± 0.1[**] |
| IR36 | 32 °C | 3.1 ± 0.4 | 183.3 ± 7.6 | 264.7 ± 13.8 | 30.7 ± 1.5 | 81.5 ± 1.8 | 20.5 ± 0.3 | 8.1 ± 0.3 | 2.5 ± 0.0 |
|  | 40 °C | 1.0 ± 0.1[**] | 85.0 ± 13.5[**] | 176.7 ± 17.6[**] | 51.4 ± 9.3[**] | 73.5 ± 1.1[**] | 16.0 ± 0.1[**] | 6.8 ± 0.2[**] | 2.3 ± 0.1[**] |

**Notes.**

[*] and [**] indicate significance differences between the control (32 °C) ang high (°C) temperature treatments (one-tailed Student's $t$-test).

[*]$P < 0.05$.

[**]$P < 0.01$.

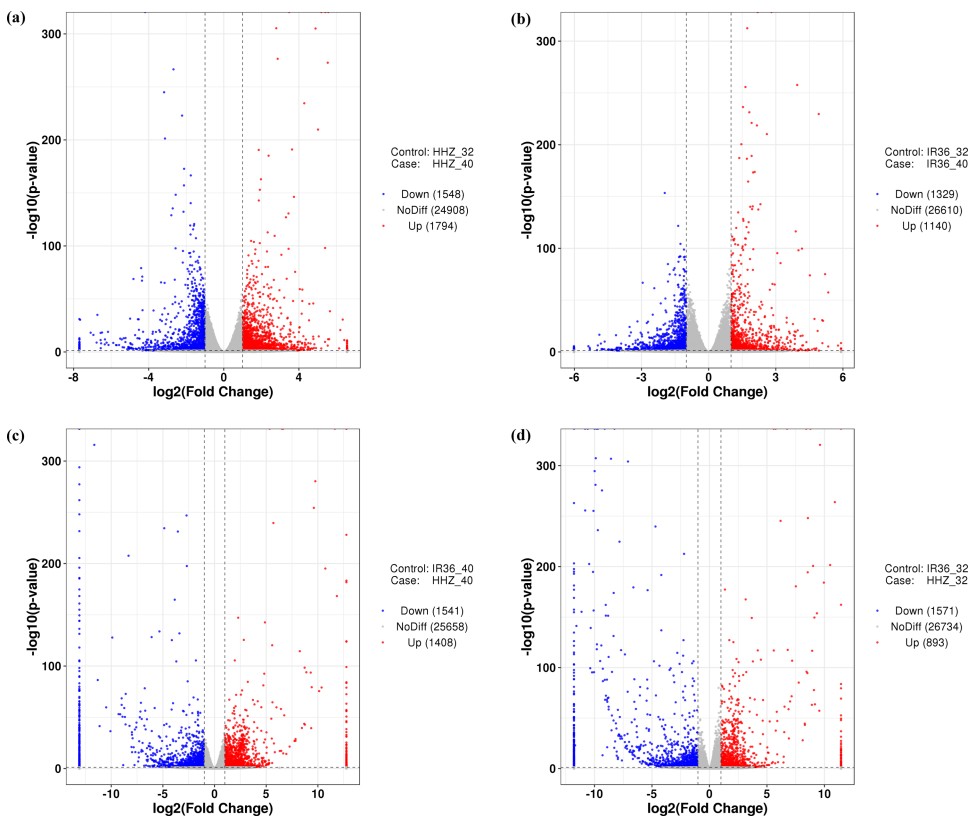

**Figure 2** **Gene expression in the four comparison groups.** (A) HHZ_32 vs. HHZ_40, (B) IR36_32 vs. IR36_40, (C) IR36_40 vs. HHZ_40, and (D) IR36_32 vs. HHZ_32. Red (upregulated) and blue (downregulated) dots indicate significant differences in gene expression, whereas gray dots represent genes with no significant difference in expression.

**Table 2** **Statistics of RNA sequencing results.**

| Sample | HHZ_32 | HHZ_40 | IR36_32 | IR36_40 |
|---|---|---|---|---|
| Raw reads | 44231722 | 45513241 | 45877838 | 46465046 |
| Clean reads | 44032896 (99.6%) | 45256701 (99.4%) | 45580821 (99.4%) | 46252929 (99.5%) |
| Total mapped | 38834391 (87.8%) | 39148950 (86.0%) | 39541858 (86.2%) | 40418126 (87.0%) |
| Uniquely mapped | 37502957 (84.8%) | 37759013 (83.0%) | 38120438 (83.1%) | 38853775 (83.1%) |
| Multiply mapped | 1331434 (3.0%) | 1389937 (3.1%) | 1421421 (3.1%) | 1561018 (3.6%) |

**Notes.**

HHZ_32: The sample of HHZ treated with 32 °C; HHZ_40: The sample of HHZ treated with 40 °C; IR36_32: The sample of IR36 treated with 32 °C; IR36_40: The sample of IR36 treated with 40 °C.

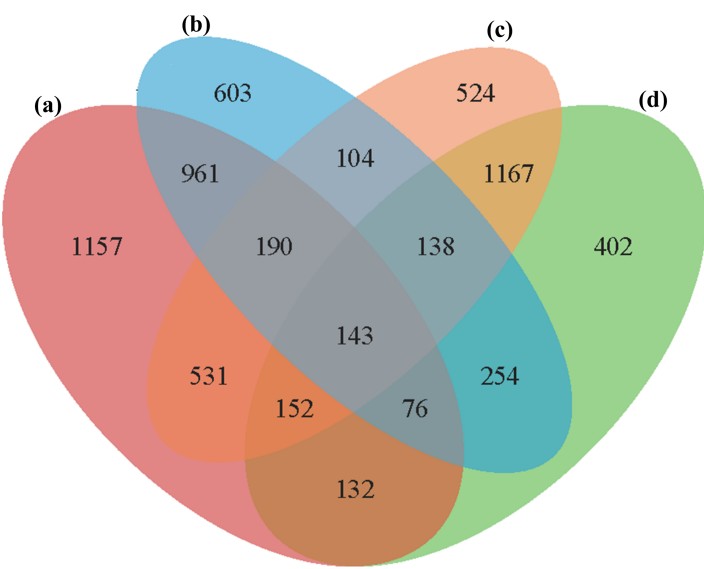

**Figure 3** **Venn diagrams for DEGs in the four comparison groups.** (A) HHZ_32 vs. HHZ_40, (B) IR36_32 vs. IR36_40, (C) IR36_40 vs. HHZ_40, and (D) IR36_32 vs. HHZ_32. 2.

**Table 3** **Classification of three categories of DEGs.**

| Categories | Subgroups | Number of DEGs |
| --- | --- | --- |
| RHR | Only HHZ_32 vs. HHZ_40 | 1,157 |
| | HHZ_32 vs. HHZ_40 ∩ IR36_40 vs. HHZ_40 | 531 |
| SHR | Only IR36_32 vs. IR36_40 | 603 |
| | IR36_32 vs. IR36_40 ∩ IR36_40 vs. HHZ_40 | 104 |
| CHR | Only IR36_40 vs. HHZ_40 | 524 |
| | HHZ_32 vs. HHZ_40 ∩ IR36_32 vs. IR36_40, HHZ_32 vs. HHZ_40 ∩ IR36_32 vs. IR36_40 ∩ IR36_40 vs. HHZ_40 | 1,151 |

**Notes.**
RHR, heat-resistant-cultivar-related genes; SHR, heat-susceptible-cultivar-related genes; CHR, common heat stress-response genes.

## Analysis of GO annotation

The purpose of the GO enrichment analysis was to obtain GO functional terms with significant enrichment of DEGs and thus reveal the possible functions of the DEGs. Of all DEGs, 2,307 (69.0%), 1,680 (68.0%), 1,832 (62.1%), and 1,472 (59.8%) DEGs were enriched in GO terms in HHZ_32 vs. HHZ_40, IR36_32 vs. IR36_40, IR36_40 vs. HHZ_40, and IR36_32 vs. HHZ_32 groups, respectively. There were 75, 11, 13, and 31 significant GO terms observed in HHZ_32 vs. HHZ_40, IR36_32 vs. IR36_40, IR36_40 vs. HHZ_40, and IR36_32 vs. HHZ_32, respectively (Fig. 4). The maximum number of DEGs was observed for the heterocycle biosynthetic process in the IR36_40 vs. HHZ_40 group. In IR36_32 vs. IR36_40 and HHZ_32 vs. HHZ_40, the DEGs were enriched in the terms response to stimulus, response to temperature stimulus, and response to heat in the biological process category. Within the cellular component category, the DEGs were commonly enriched in the terms chromatin, DNA packaging complex, and nucleosome in the IR36_32 vs.
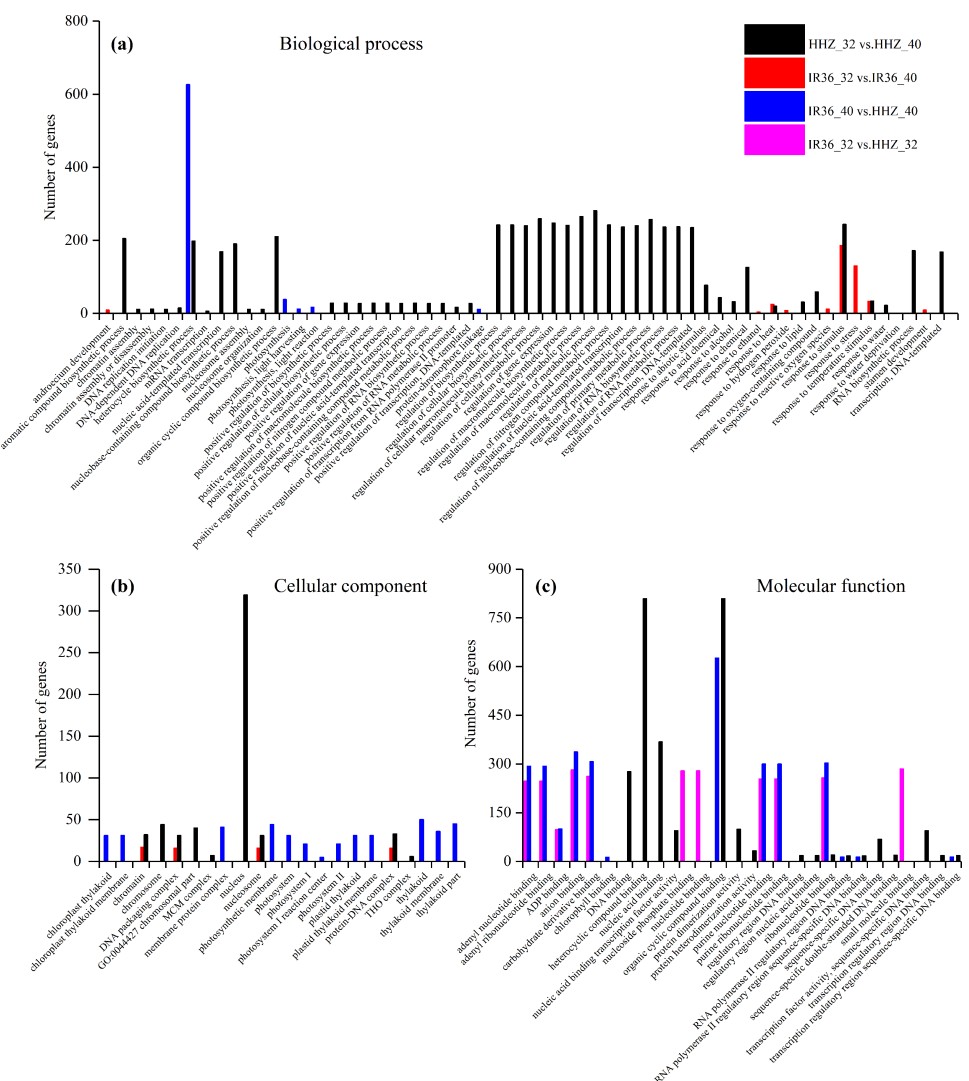

**Figure 4** **Enriched GO terms ($P < 0.05$) of all DEGs.** (A) biological process, (B) cellular component, and (C) molecular function.

IR36_40 and HHZ_32 vs. HHZ_40 groups. However, there were no common GO terms in the category of molecular function in the IR36_32 vs. IR36_40 and HHZ_32 vs. HHZ_40 groups.

We further identified GO term categories for DEGs in the RHR, SHR, and CHR categories (Fig. 5 and Table S5). Among the 1,689 DEGs in RHR, 54 significant GO terms were detected. However, no significant GO terms were observed among the 707 DEGs in SHR. In CHR, 30 significant GO terms were detected. In the CHR group, eight significant GO terms were observed in the biological process category, including response to stimulus, response to temperature stimulus, and response to heat; 17 GO terms were in the cellular component category; and two significant GO terms were in the molecular function category. In the RHR group, 30, 14 and 10 significant GO terms were in the

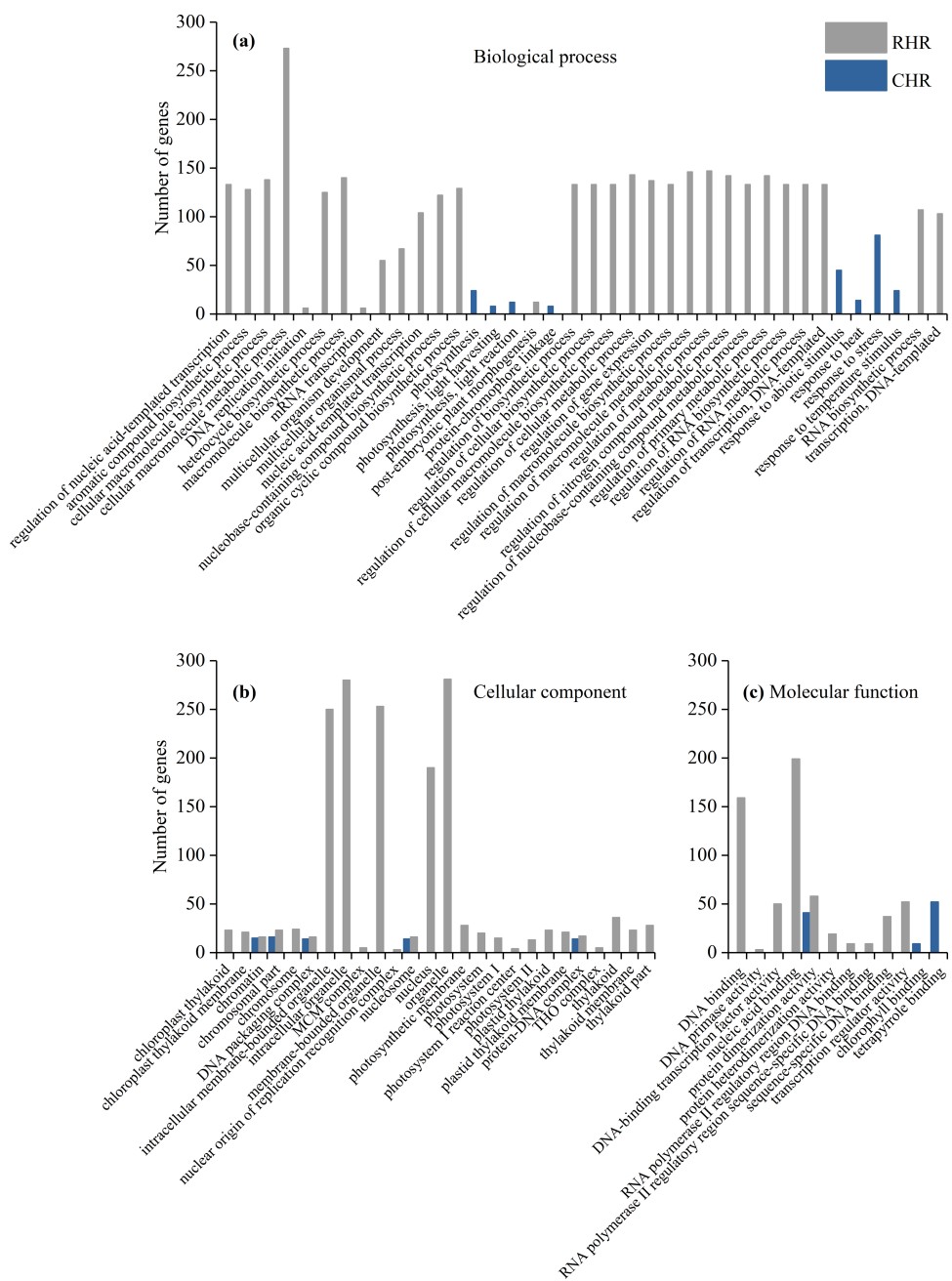

**Figure 5  Enriched GO terms ( *P* < 0.05) of DEGs in RHR and CHR.** (A) biological process, (B) cellular component, and (C) molecular function.

biological process, cellular component, and molecular function categories, respectively. The most significant GO terms, in decreasing order, were RNA biosynthetic process, nucleus, and DNA binding. In the molecular function category, 50 DEGs were specifically assigned to DNA-binding transcription factor activity, which may play an important role in heat stress tolerance.
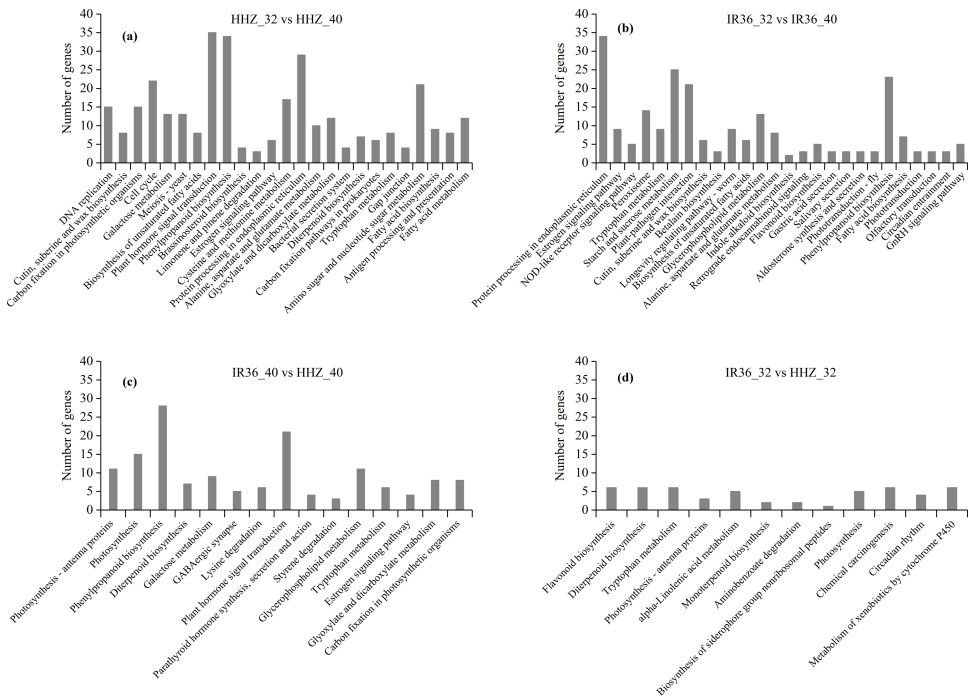

**Figure 6** **KEGG enrichment analysis of all DEGs.** (A) HHZ_32 vs. HHZ_40, (B) IR36_32 vs. IR36_40, (C) IR36_40 vs. HHZ_40, and (D) IR36_32 vs. HHZ_32.

The 50 DEGs of DNA-binding transcription factor activity could be divided into 11 transcription factor (TF) families, including HSF (1), WRKY (6), MADS (12), HD-ZIP (7), GATA (3), ERF (12), ABAI (1), b-ZIP (4), ARR-B (2), E2F (1), and NF-YA (1). Expression of the genes *BGIOSGA006348* of HSF, *BGIOSGA010835* of ABAI, *BGIOSGA010142* of HAP, and *BGIOSGA000303* and *BGIOSGA000304* of ARR-B was significantly upregulated. In addition, five genes in WRKY, eight genes in MADS, two genes in HD-ZIP, two genes in GATA, six genes in ERF, and two genes in b-ZIP were also upregulated (Table S6). These results suggest that, these 30 TF genes may play important roles in heat stress resistance.

## Analysis of KEGG pathway enrichment

In the KEGG analysis, 1,158 DEGs were classified into 225, 191, 239, and 211 functional pathways in HHZ_32 vs. HHZ_40; 838 DEGs in IR36_32 vs. IR36_40; 732 DEGs in IR36_40 vs. HHZ_40; and 539 DEGs in IR36_32 vs. HHZ_32. A total of 79 pathways were significant (*P-value* < 0.05) (Fig. 6). Among these pathways, the phenylpropanoid biosynthesis pathway was common in HHZ_32 vs. HHZ_40, IR36_32 vs. IR36_40, and IR36_40 vs. HHZ_40, which suggests that heat stress impaired phenylpropanoid biosynthesis.

Based on further analysis of the three categories with different heat-stress responses, 146 DEGs in RHR were involved in 15 overrepresented pathways, including purine metabolism, pyrimidine metabolism, and amino sugar and nucleotide sugar metabolism; 45 DEGs in SHR were involved in 11 overrepresented pathways, including arginine biosynthesis, starch

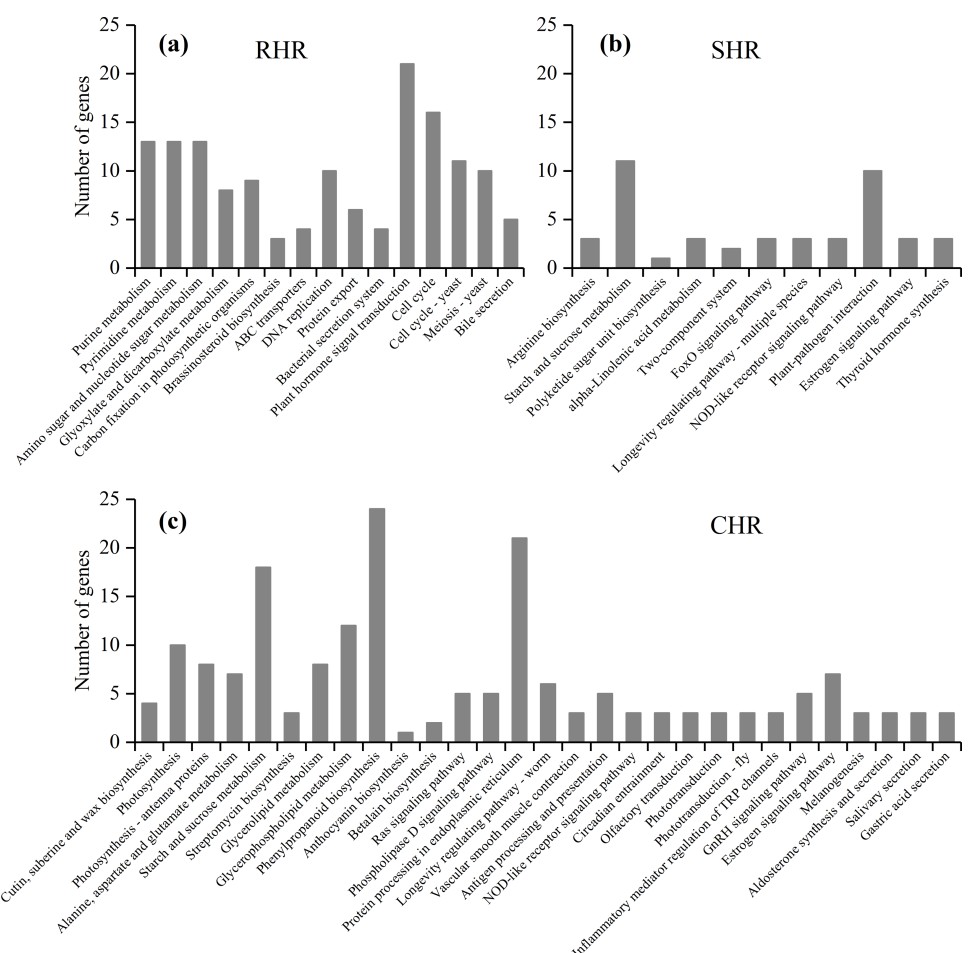

**Figure 7** **KEGG enrichment analysis for heat stress responsive genes from the three categories.** (A) RHR, (B) SHR, and (C) CHR.

and sucrose metabolism, and polyketide sugar unit biosynthesis; and 184 DEGs in CHR were involved in 29 overrepresented pathways (Fig. 7 and Table S7).

A previous study showed that plant hormones are important for panicle development. Among the 15 KEGG pathways in RHR, 21 DEGs were involved in plant hormone signal transduction, of which 14 DEGs were upregulated in HHZ; three DEGs were involved in cytochrome P450 metabolism, which plays a role in brassinosteroid (BR) biosynthesis; and two were upregulated (Table 4).

In SHR and CHR, there were three common pathways: the starch and sucrose metabolism pathway, the NOD-like receptor signaling pathway, and the estrogen signaling pathway. Carbohydrate accumulation was essential for panicle development. In the KEGG analysis, seven DEGs involved in starch and sucrose metabolism were observed in SHR and 18 DEGs involved in starch and sucrose metabolism were observed in CHR. In SHR, the genes in HHZ were not different between HHZ_40 and HHZ_32. However, genes *BGIOSGA010570* and *BGIOSGA026140* encoding sucrose synthase (EC 2.4.1.13), genes *BGIOSGA026976*,

Wang et al. (2019), *PeerJ*, DOI 10.7717/peerj.7595

**Table 4 Gene expression of DEGs in Plant hormone signal transduction and BR biosynthesis of RHR.**

| ID | Gene annotation | Cultivar | baseMean | 32 °C | 40 °C | log2FoldChange | pval |
|---|---|---|---|---|---|---|---|
| BGIOSGA018672 | Pseudo histidine-containing phosphotransfer protein 2 | HHZ | 64.7 | 36.9 | 92.5 | 1.33 | 0.00 |
| | | IR36 | 59.4 | 49.2 | 69.7 | 0.50 | 0.06 |
| BGIOSGA004140 | Probable protein phosphatase 2C 8 | HHZ | 665.2 | 219.8 | 1110.7 | 2.34 | 0.00 |
| | | IR36 | 465.6 | 355.3 | 575.9 | 0.70 | 0.00 |
| BGIOSGA005312 | Two-component response regulator ORR3 | HHZ | 50.6 | 28.6 | 72.6 | 1.35 | 0.00 |
| | | IR36 | 26.3 | 24.2 | 28.5 | 0.24 | 0.58 |
| BGIOSGA024710 | Auxin-responsive protein IAA24 | HHZ | 807.3 | 458.2 | 1156.3 | 1.34 | 0.00 |
| | | IR36 | 828.1 | 653.4 | 1002.7 | 0.62 | 0.00 |
| BGIOSGA010835 | ABSCISIC ACID-INSENSITIVE 5-like protein 2 | HHZ | 146.5 | 85.8 | 207.2 | 1.27 | 0.00 |
| | | IR36 | 83.4 | 74.0 | 92.8 | 0.33 | 0.26 |
| BGIOSGA011032 | Probable protein phosphatase 2C 30 | HHZ | 102.9 | 53.4 | 152.4 | 1.51 | 0.00 |
| | | IR36 | 113.6 | 108.0 | 119.3 | 0.14 | 0.69 |
| BGIOSGA015611 | Probable protein phosphatase 2C 37 | HHZ | 86.1 | 44.3 | 127.8 | 1.53 | 0.00 |
| | | IR36 | 75.8 | 52.0 | 99.6 | 0.94 | 0.00 |
| BGIOSGA019301 | Auxin-responsive protein IAA16 | HHZ | 97.4 | 56.6 | 138.3 | 1.29 | 0.00 |
| | | IR36 | 79.7 | 76.6 | 82.8 | 0.11 | 0.61 |
| BGIOSGA008704 | Auxin-responsive protein SAUR36 | HHZ | 36.6 | 22.9 | 50.3 | 1.14 | 0.00 |
| | | IR36 | 23.1 | 22.5 | 23.6 | 0.07 | 0.91 |
| BGIOSGA012535 | ARATH Protein ETHYLENE INSENSITIVE 3 | HHZ | 2890.1 | 1268.4 | 4511.8 | 1.83 | 0.00 |
| | | IR36 | 2293.2 | 1543.8 | 3042.6 | 0.98 | 0.00 |
| BGIOSGA037772 | ARATH Transcription factor PIF1 | HHZ | 27.5 | 14.2 | 40.8 | 1.52 | 0.00 |
| | | IR36 | 17.3 | 13.3 | 21.4 | 0.69 | 0.24 |
| BGIOSGA000304 | Two-component response regulator ORR26 | HHZ | 143.8 | 92.5 | 195.0 | 1.08 | 0.00 |
| | | IR36 | 110.4 | 85.9 | 134.9 | 0.65 | 0.00 |
| BGIOSGA004789 | Probable protein phosphatase 2C | HHZ | 522.2 | 301.9 | 742.5 | 1.30 | 0.00 |
| | | IR36 | 623.2 | 501.8 | 744.5 | 0.57 | 0.02 |
| BGIOSGA037837 | Auxin-responsive protein SAUR72 | HHZ | 3.5 | 1.0 | 6.0 | 2.55 | 0.04 |
| | | IR36 | 0.8 | 1.3 | 0.3 | −2.07 | 0.67 |
| BGIOSGA024374 | Two-component response regulator ORR7 | HHZ | 17.8 | 29.5 | 6.2 | −2.26 | 0.00 |
| | | IR36 | 49.3 | 58.2 | 40.4 | −0.53 | 0.10 |
| BGIOSGA036617 | Transcription factor TGAL11 | HHZ | 308.0 | 423.3 | 192.8 | −1.13 | 0.00 |
| | | IR36 | 572.4 | 700.8 | 444.0 | −0.66 | 0.00 |

Wang et al. (2019), *PeerJ*, DOI 10.7717/peerj.7595

**Table 4** (*continued*)

| ID | Gene annotation | Cultivar | baseMean | 32 °C | 40 °C | log2FoldChange | pval |
|---|---|---|---|---|---|---|---|
| BGIOSGA034772 | BTB/POZ domain and ankyrin repeat-containing protein NH5.1 | HHZ | 1148.1 | 1629.2 | 667.0 | −1.29 | 0.00 |
| | | IR36 | 1335.2 | 1698.8 | 971.6 | −0.81 | 0.00 |
| BGIOSGA010559 | Protein TIFY 10a | HHZ | 339.2 | 465.1 | 213.4 | −1.12 | 0.00 |
| | | IR36 | 374.0 | 492.0 | 256.0 | −0.94 | 0.00 |
| BGIOSGA010919 | Abscisic acid receptor PYL5 | HHZ | 34.8 | 55.0 | 14.5 | −1.92 | 0.00 |
| | | IR36 | 52.9 | 58.0 | 47.8 | −0.28 | 0.33 |
| BGIOSGA023368 | Two-component response regulator ORR25 | HHZ | 4.4 | 8.8 | 0.0 | −Inf | 0.00 |
| | | IR36 | 3.2 | 5.5 | 1.0 | −2.49 | 0.15 |
| BGIOSGA034767 | BTB/POZ domain and ankyrin repeat-containing protein NH5.2 | HHZ | 1147.8 | 1623.6 | 672.0 | −1.27 | 0.00 |
| | | IR36 | 1304.2 | 1737.3 | 871.0 | −1.00 | 0.00 |
| BGIOSGA002945 | Cytochrome P450 90D2 | HHZ | 178.2 | 118.0 | 238.4 | 1.01 | 0.00 |
| | | IR36 | 184.9 | 164.3 | 205.5 | 0.32 | 0.05 |
| BGIOSGA014915 | Cytochrome P450 724B1 | HHZ | 1872.9 | 2570.7 | 1175.2 | −1.13 | 0.00 |
| | | IR36 | 1251.4 | 1482.6 | 1020.2 | −0.54 | 0.00 |
| BGIOSGA001585 | Cytochrome P450 734A6 | HHZ | 123.1 | 178.3 | 67.9 | −1.39 | 0.00 |
| | | IR36 | 202.3 | 267.6 | 136.9 | −0.97 | 0.00 |

*BGIOSGA009181*, and *BGIOSGA030796* encoding trehalose-6-phosphate synthase (EC 2.4.1.15), and gene *BGIOSGA000509* encoding trehalose-6-phosphate phosphatase (EC 3.1.3.12) were significantly downregulated in IR36_40 compared with IR36_32. However, gene *BGIOSGA031385* encoding beta-amylase (EC 3.2.1.2) was significantly upregulated in IR36_40 compared with IR36_32 (Table 5).

## qRT-PCR verification

To confirm the accuracy of the RNA-Seq results, ten representative DEGs each from the HHZ_32 vs. HHZ_40 (a) and IR36_32 vs. IR36_40 (b) groups, as well as five DEGs each from the IR36_40 vs. HHZ_40 (c) and IR36_32 vs. HHZ_32 (d) groups were chosen to determine relative expression. Of the ten DEGs from the HHZ_32 vs. HHZ_40 group, five were in RHR: *BGIOSGA022020* is related to BR synthesis, *BGIOSGA006348* encodes a heat shock factor (Hsf), *BGIOSGA017088* is involved in the ETH TF family, *BGIOSGA006285* participates in ethylene responsive regulation, and *BGIOSGA024710* is an auxin-responsive gene involved in plant hormone transduction. Among the ten DEGs from the IR36_32 vs. IR36_40 group, five were in SHR and encoded cytokinin oxidase/dehydrogenase (*BGIOSGA005140*), sucrose synthase (*BGIOSGA026140*), trehalose-6-phosphate synthase (*BGIOSGA026976*), trehalose-6-phosphate phosphatase (*BGIOSGA000509*), and catalase (*BGIOSGA007252*). Four DEGs were in CHR from the HHZ_32 vs. HHZ_40 and IR36_32 vs. IR36_40 groups and two common genes, *BGIOSGA032653* and *BGIOSGA015767*, were validated. *BGIOSGA032653* is involved in phenylpropanoid biosynthesis and *BGIOSGA015676* encodes a heat shock protein (HSP). The qRT-PCR results for the DEGs were all consistent with the RNA-Seq data (Fig. 8).

## DISCUSSION

The exposure of rice plants to high temperature growing conditions during spikelet differentiation inhibited panicle initiation and reduced spikelet number per panicle (Fig. 1). Previous studies have shown that the genes *SP1*, *ASP1*, *TUT1*, *PAA2*, and *OsALMT7* are closely related to branch and spikelet development in rice (*Bai et al., 2015*; *Heng et al., 2018*; *Li et al., 2010*). However, in the current study, we observed no significant difference in the expression of these genes between the 40 °C treatment and the 32 °C control treatment in either rice cultivar, which indicates that the expression of these genes might not be inhibited in young panicles exposed to high temperature.

In general, the upregulation of HSPs contributes to the heat stress response in plants (*Guan et al., 2010*; *Jagadish et al., 2010*; *Jung et al., 2013*). *Moon et al. (2014)* reported that heterologous overexpression of *OsHSP1* (*BGIOSGA015767*, encoding a HSP) increased heat tolerance in Arabidopsis. However, in the current study, *BGIOSGA15767* expression was upregulated in both HHZ (log2 (HHZ_40/HHZ_32) = 5.7, *P*-value = 0) and IR36 (log2 (IR36_40/IR36_32) = 5.0, *P*-value = 0). In addition, there was no gene expression difference in the GO term of HSPs between cultivars, which demonstrates that the heat stress reaction is common to both rice cultivars when exposed to high temperature. The GO enrichment analysis revealed that the DEGs for the CHR group were commonly enriched in response to GO terms representing heat, stress, and temperature stimuli in the biological

**Table 5  Gene expression of DEGs in starch and sucrose metabolism in SHR.**

| ID | Gene annotation | Cultivar | baseMean | IR36_32 | IR36_40 | log2FoldChange | *P*-value |
|---|---|---|---|---|---|---|---|
| BGIOSGA010570 | Sucrose synthase | HHZ | 14318.1 | 18545.9 | 10090.4 | −0.88 | 0.00 |
| | | IR36 | 13352.6 | 18616.3 | 8088.8 | −1.20 | 0.00 |
| BGIOSGA026140 | Sucrose synthase | HHZ | 13.5 | 16.5 | 10.4 | −0.67 | 0.24 |
| | | IR36 | 16.8 | 24.7 | 8.9 | −1.47 | 0.01 |
| BGIOSGA026976 | trehalose-6-phosphate synthase, putative, expressed | HHZ | 651.9 | 572.0 | 731.7 | 0.36 | 0.12 |
| | | IR36 | 691.3 | 399.7 | 982.9 | 1.30 | 0.00 |
| BGIOSGA009181 | trehalose-6-phosphate synthase, putative, expressed | HHZ | 731.2 | 562.7 | 899.8 | 0.68 | 0.01 |
| | | IR36 | 988.8 | 578.3 | 1399.3 | 1.28 | 0.00 |
| BGIOSGA030796 | trehalose-6-phosphate synthase, putative, expressed | HHZ | 2.2 | 1.9 | 2.4 | 0.38 | 0.98 |
| | | IR36 | 4.3 | 0.0 | 8.6 | Inf | 0.00 |
| BGIOSGA000509 | Trehalose-6-phosphate phosphatase | HHZ | 175.0 | 229.9 | 120.2 | −0.94 | 0.00 |
| | | IR36 | 179.2 | 268.6 | 89.9 | −1.58 | 0.00 |
| BGIOSGA031385 | beta-amylase, putative, expressed | HHZ | 19.3 | 17.7 | 20.8 | 0.23 | 0.65 |
| | | IR36 | 28.0 | 17.4 | 38.5 | 1.15 | 0.01 |

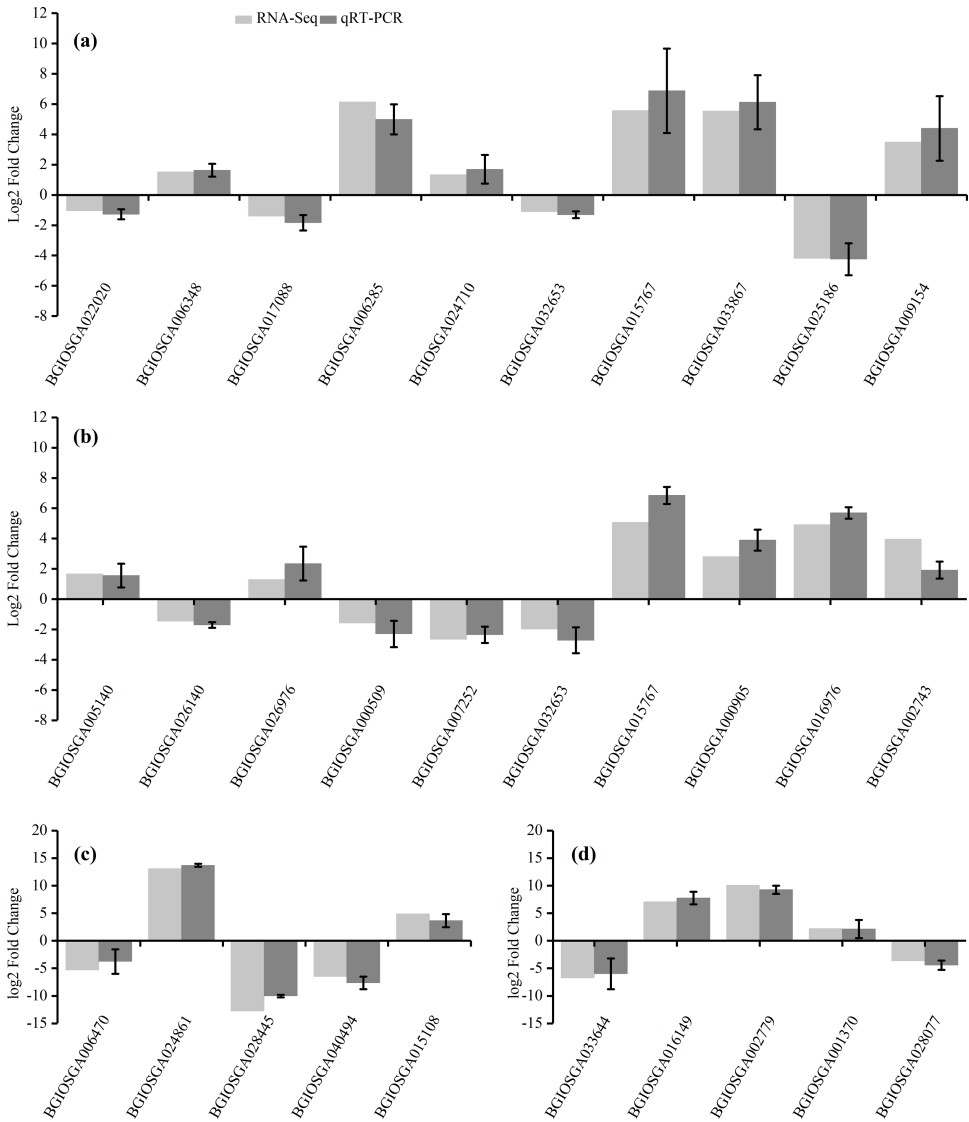

**Figure 8** **Gene expression levels determined by RNA-Seq and qRT-PCR.** (A) HHZ_32 vs. HHZ_40, (B) IR36_32 vs. IR36_40, (C) IR36_40 vs. HHZ_40, and (D) IR36_32 vs. HHZ_32.

process category (Fig. 5). These results demonstrate that the heat stress response did not directly inhibit panicle development but rather may disrupt physiological processes related to panicle development.

An important factor determining heat tolerance is antioxidant capacity (*Lan et al., 2016*). *Buer, Imin & Djordjevic (2010)* reported that flavonoids can positively regulate reactive oxygen species (ROS), which can affect the transport of plant hormones and influence pollen development. The flavonoid synthesis pathway was overrepresented in the IR36_32 vs. IR36_40 group. Specifically, five genes involved in flavonoid synthesis were downregulated at 40 °C, which might indicate a reduction in the antioxidant capacity of IR36 under heat stress. In addition, 14 DEGs in the IR36_32 vs. IR36_40 group were enriched in

the peroxisome pathway. Among these, 10 DEGs were significantly downregulated and four DEGs were significantly upregulated. However, the peroxisome pathway was not significant in the KEGG analysis of HHZ_32 vs. HHZ_40 (Fig. 6). BGIOSGA007252 and BGIOSGA011520, which encode catalase (EC:1.11.1.6), were significantly downregulated in IR36 at 40 °C compared with 32 °C, whereas no expression differences were observed in HHZ_32 vs. HHZ_40. This finding suggests that high temperature had a greater negative effect on the antioxidant capacity of IR36 than of HHZ, which provides a primary explanation for the greater heat injury observed in the young IR36 panicles than in those of HHZ.

Regulation of endogenous hormones affects the development of young panicles. *Wu et al. (2017)* reported that a lower spikelet number under high temperature growing conditions was associated with cytokinin degradation. In the current study, BGIOSGA001314, which encodes a cytokinin-activity enzyme, did not differ between the 40 °C and 32 °C treatments in HHZ (log2 (HHZ_40/HHZ_32) = −0.41) or IR36 (log2 (IR36_40/IR36_32) = −0.38). However, the gene BGIOSGA005140, which encodes cytokinin oxidase/dehydrogenase, was significantly upregulated in the IR36_32 vs. IR36_40 group (log2 fold change = 1.67, *P-value* = 0.004), but was not different in the HHZ_32 vs. HHZ_40 group (log2 fold change = 0.86, *P-value* = 0.088). These results are consistent with those of *Wu et al. (2016)* and suggest that spikelet formation is associated with cytokinin degradation and greater degradation occurs at high temperatures in the heat-susceptible cultivar than in the heat-tolerant cultivar.

The DEGs in RHR were enriched in 54 GO terms (Fig. 5). The GO term analysis revealed biological processes promoting resistance to heat stress in the heat-tolerant cultivar HHZ. Downregulation of *BGIOSGA022020* in the heterocycle biosynthetic process induces GRAS protein reduction, which promotes BR synthesis to enhance heat tolerance (*Vriet, Russinova & Reuzeau, 2012*). In the molecular function category for RHR, 50 DEGs were involved in DNA-binding transcription factor activity. *BGIOSGA006348* encoded an HSF TF and was upregulated in the HHZ_32 vs. HHZ_40 group, although differences were not observed in the IR36_32 vs. IR36_40 group. *Wang, Qian & Shou (2009)* reported that the higher expression of heat shock TFs contributed to high temperature tolerance. WRKY genes encode TFs that play important roles in abiotic stress responses (*Chen et al., 2010*), especially to abscisic acid (ABA) (*Zhen et al., 2005*). In this study, six DEGs were WRKY TFs, namely, *BGIOSGA003134*, *BGIOSGA017063*, *BGIOSGA029574*, *BGIOSGA005924*, *BGIOSGA024948*, and *BGIOSGA033505*, which might promote young panicle development associated with sucrose consumption mediated by ABA under high temperature (*Feng et al., 2018*). However, few studies have reported the relationship between the WRKY family and heat resistance, which should be further studied. *BGIOSGA029574* is a general stress-response gene, which has putative functions in distinct cellular processes, such as transcription regulation, stress response, and sugar metabolism under Fe-excess-induced, dark-induced, and drought-induced stress (*Ricachenevsky et al., 2010*). Among the six WRKY genes, BGIOSGA017063 was downregulated while the other five genes were upregulated, although the gene has not been cloned for the gene function analysis and therefore requires further study. Of the 10 DEGs in the ETH family, five genes were

downregulated and the downregulation of *BGIOSGA017088* reduced the ABA content and promoted gibberellin (GA) signal transduction, which is beneficial for rice plant growth (*Yaish et al., 2010*). The upregulation of *BGIOSGA006285*, *BGIOSGA010867*, *BGIOSGA030019*, *BGIOSGA005915*, and *BGIOSGA012535* plays an important role in ethylene response regulation. *Cao et al. (2006)* reported that the upregulation of *BGIOSGA005915* enhanced tolerance to salt, cold, drought, and wounding, and the current study reveals that this gene might also contribute to the improvement of high-temperature stress resistance. *BGIOSGA000303* and *BGIOSGA000304* are genes in the cytokinin receptor family, and the upregulation of these two genes promotes cytokinin activation (*Ito & Kurata, 2006*). The MADs box gene is related to flower development (*Kobayashi et al., 2012*) and the upregulation of the MAD genes in RHR indicated that the MAD family might enhance heat stress tolerance. The HZ-ZIP TF family might have a similar function.

In the RHR category, the DEGs enriched in the KEGG pathways appear beneficial for heat-stress tolerance, including plant hormone signal transduction and BR biosynthesis. Twenty-one DEGs were involved in plant hormone signal transduction, of which 14 DEGs were upregulated, including the auxin-responsive genes *BGIOSGA024710*, *BGIOSGA001585*, *BGIOSGA019301*, and *BGIOSGA037837*, which facilitate rice plant growth (*Hagen & Guilfoyle, 2002*). In BR biosynthesis, *BGIOSGA002945,* which encodes *D2/CYP90D2*, a gene that catalyzes the steps from 6-deoxoteasterone to 3-dehydro-6-deoxoteasterone and from teasterone to 3-dehydroteasterone, was upregulated to promote BR synthesis in the latter pathway (*Hong et al., 2003*), and *BGIOSGA001585* was downregulated to promote BR activity (*Sakamoto et al., 2011*). The genes related to hormone signal transduction and BR biosynthesis might contribute to young panicle development under high temperature. Seven DEGs involved in plant hormone signal transduction were downregulated, and among these, *BGIOSGA036617*, *BGIOSGA034767*, and *BGIOSGA010559* have not been cloned for functional analysis while *BGIOSGA034772* plays a more important role in organismal development. The genes *BGIOSGA024374*, *BGIOSGA023368*, *BGIOSGA000304* and *BGIOSGA005312* are A-type response regulated genes (*Jain, Tyagi & Khurana, 2006*). However, it is unclear whether the downregulation of *BGIOSGA024374* and *BGIOSGA023368* contributes to improved heat tolerance in rice varieties. In addition, the downregulated gene, *BGIOSGA010919*, is an ABA receptor. *Tian et al. (2015)* reported that ABA accumulation upregulates gene expression. In the current study, downregulation of *BGIOSGA010919* may contribute to excessive ABA accumulation. The role of ABA in panicle development requires further study. *BGIOSGA014915*, which participates in BR synthesis, was downregulated in RHR. Previous reports have found that BRs can modulate the metabolic responses of plants to abiotic environmental stresses (*Vriet, Russinova & Reuzeau, 2012*; *Wang, Zhang & Yang, 2018*). BR accumulation reportedly reduces spikelet degeneration under nitrogen application (*Zhang, 2018*). *BGIOSGA002945* and *BGIOSGA014915* participate in different BR biosynthesis pathways (*Shi, 2015*), but *BGIOSGA002945* may play a more important role in modulating spikelet development under high temperature.

Carbohydrate storage and utilization are essential for panicle initiation (*Tian et al., 2016*). The KEGG analysis showed that the phenylpropanoid biosynthesis pathway was commonly overrepresented in HHZ_32 vs. HHZ_40, IR36_32 vs. IR36_40, and IR36_40 vs. HHZ_40. The phenylpropanoid biosynthesis pathway is involved in lignin synthesis, which suggests that high temperature inhibits lignin synthesis; however, phenylpropanoid biosynthesis was not associated with heat tolerance in our heat resistant cultivar (Fig. 6). In the SHR category, seven DEGs were enriched in the starch and sucrose metabolism pathway (Fig. 7B). The gene *BGIOSGA031385*, which encodes beta-amylase, was significantly upregulated in IR36_32 vs. IR36_40, suggesting that it promoted starch hydrolysis and reduced carbohydrate storage. The genes *BGIOSGA010570* and *BGIOSGA026140*, which encode sucrose synthesis, were significantly downregulated in the IR36_32 vs. IR36_40 group, whereas no difference in expression was observed in the HHZ_32 vs HHZ_40 group. Sucrose degrades into uridine 5′-diphosphoglucose and fructose, which are major forms of carbon used for energy. Impairment of sucrose synthase activity reportedly reduced resistance to heat stress (*Hirose, Scofield & Terao, 2008*; *Takehara et al., 2018*). The results of the current study suggest that impaired carbohydrate metabolism in the heat-susceptible cultivar aggravated spikelet reduction. The starch and sucrose pathway genes were also highly represented in the CHR group (Fig. 7C). Such genes are involved in the downregulation of genes encoding beta-fructofuranosidase, fructokinase, beta-glucosidase, trehalose-6-phosphate phosphatase, alpha-trehalase, and others. Trehalose-6-phosphate synthase, trehalose-6-phosphate phosphatase, and alpha-trehalase are involved in trehalose synthesis. Trehalose plays an important role in abiotic stress resistance, and trehalose-6-phosphate, an intermediate product of trehalose synthesis participates in sucrose signal transduction (*Lunn et al., 2006*; *Ruan, 2014*). *Nunes & Paul (2013)* reported that trehalose-6-phosphate served as a sugar signal that could induce the expression of genes associated with the alleviation of abiotic stress injury. In this study, certain DEGs in the CHR group were also upregulated to promote trehalose-6-phosphate synthesis, and the upregulation of *BGIOSGA026976*, *BGIOSGA009181*, and *BGIOSGA030796* promoted trehalose-6-phosphate synthesis in SHR. These findings indicate that trehalose-6-phosphate synthesis may be a normal response of young rice panicles to high temperature and that the heat-sensitive rice cultivar synthesizes trehalose-6-phosphate more readily than the heat-tolerant cultivar in response to heat stress. However, the gene encoding trehalose-6-phosphate phosphatase, *BGIOSGA000509*, was significantly downregulated in IR36 at 40 °C compared with that at 32 °C, which might cause a decrease in trehalose content and in turn disrupt carbohydrate distribution. Our results suggest that trehalose-6-phosphate metabolism was disordered under the high temperature condition and that the effects were more severe in the heat-susceptible cultivar than in the heat-tolerant cultivar.

A close relationship is observed between endogenous hormones and carbohydrate accumulation, which may suggest that the regulation of endogenous hormones in heat-tolerant varieties promotes carbohydrate utilization. The identification of DEGs in this study could improve understanding of the molecular mechanisms of heat resistance in young panicles. In the practice of rice production and breeding, DEGs associated with hormone metabolism in the RHR category and DEGs associated with starch and

metabolism in the SHR category under high temperature could be used to quickly identify heat tolerant cultivars.

## CONCLUSIONS

In summary, heat stress-responsive DEGs in young panicles were identified by a transcriptome analysis of a heat-tolerant rice cultivar and a heat-susceptible rice cultivar grown at high temperature (40 °C) and a control temperature (32 °C). The statistical analysis of 5,533 DEGs revealed three categories of genes (RHR, SHR, and CHR) containing a total of 4,070 DEGs. We highlighted the differential expression of a group of DNA-binding TFs that was significantly enriched in the RHR category as well as the differential expression of genes involved in the starch and sucrose metabolism pathway that were overrepresented in the SHR category. Overall, DEGs related to plant hormones and signal transduction might be specifically beneficial for young panicle development at high temperature. Heat-tolerant cultivars seem to increase endogenous hormones and maintain a stable carbohydrate metabolism pathway under high temperature. However, certain metabolic pathways, including starch and sucrose metabolism, were much more damaged in the heat susceptible cultivars under high temperatures, and this damage might have inhibited the panicle development.

### Funding

This study was funded by the National Key Research and Development Program of China (2017YFD0300409) and the Special Fund for China Agricultural Research System (CARS-01-07B). The funders had no role in study design, data collection and analysis, decision to publish, or preparation of the manuscript.

### Grant Disclosures

The following grant information was disclosed by the authors:
National Key Research and Development Program of China: 2017YFD0300409.
Special Fund for China Agricultural Research System: CARS-01-07B.

### Competing Interests

The authors declare there are no competing interests.

### Author Contributions

- Yaliang Wang conceived and designed the experiments, performed the experiments, analyzed the data, prepared figures and/or tables, authored or reviewed drafts of the paper, approved the final draft.
- Yikai Zhang prepared figures and/or tables, authored or reviewed drafts of the paper.
- Qiang Zhang analyzed the data, authored or reviewed drafts of the paper.
- Yongtao Cui, Jing Xiang, Huizhe Chen, Guohui Hu, Yanhua Chen and Xiaodan Wang help to collect samples.

- Defeng Zhu and Yuping Zhang conceived and designed the experiments, contributed reagents/materials/analysis tools.

## Data Availability

The raw RNA sequence data is available in the NCBI Sequence Read Archive with accession number PRJNA508820.

## Supplemental Information

Supplemental information for this article can be found online at http://dx.doi.org/10.7717/peerj.7595#supplemental-information.

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
