# Peer review of "Comparative transcriptome analysis of panicle development under heat stress in two rice (Oryza sativa L.) cultivars differing in heat tolerance"

_PeerJ, doi:10.7717/peerj.7595_

## Round 0.1 · original submission · Major Revisions

Dear authors,

Three reviewers provided their comments and all suggested major revisions. I agree with most of their comments. In my opinion, the results and and discussion are too general, which gave readers an impression of lack of novelty. I would suggest authors dig in and put out some interesting findings. For example, you have so many DEGs, there should be difference among them in term of the magnitude of differentiation. Authors can list some DEGs that have the greatest differentiation and give some discussion. In Fig 9, it looks like most don't have significant difference, but they should be different based on what you described in your text. In line 249, what does it suggest for 143 common DEGs across four groups? In line 254, what do those DEGs in CHR represent? Please see more comments from our reviewers as following.

Once again, thank you for submitting your manuscript to PeerJ and I look forward to your revision very soon!

Reviewer 1 ·

Basic reporting

literature citation and background knowledge is enough, paper structure fits requirement. English in the manuscript need to improve. Writing skills need to improve, particularly the description of the results seems like the notes. Suggesting to use tables rather than just text description.

Experimental design

experimental design is pretty straightforward in this research, but the description is not clear. For example, authors indicated the rice plants were treated in high temperature for 9 d, but they didn't mention the stage of the plants they treated. For analysis methods of differential gene expression and GO, KEGG analysis, the description is too brief and thus is lack of sufficient information.

Validity of the findings

Findings in this research are pretty general and lacking of novelty. The pathways that were indicated to be related to heat response were also very general.

Additional comments

The manuscript need a major revision for accepting to publish.

Reviewer 2 ·

Basic reporting

In the submitted manuscript “Transcriptome comparison analysis of panicle development under heat stress in two rice cultivars differing in heat tolerance (Oryza sativa L.)” (#33492) by Wang et al. used high throughput method to identify differentially expressed genes (DEGs) in developing young panicles of heat-resistant (HHZ) and heat-susceptible (IR36) rice cultivars under heat stress. They identified 4070 heat stress-responsive genes including 1688 heat-resistant-cultivar-related genes (RHR), 707 heat-susceptible-cultivar-related genes (SHR) and 1675 common heat stress-responsive genes (CHR), and they validated 30 of them by qRT-PCR analysis and identified some DEGs and pathways involved in heat stress resistance through GO enrichment and KEGG analysis. Although the manuscript presents some interesting results, the novelty of the findings is not clear and some important deficiencies in the manuscript must be revised before publication. In addition, the manuscript writing requires a huge number of changes for both clarity and proper English usage such as in line 43-53, 61-62, 76-77, 122-123, 128, 155-159, 266-269, 297-299…

Experimental design

no comment

Validity of the findings

no comment

Additional comments

Specific comments:
- In the INTRODUCTION section, what does IPCC stand for (line 44)? Authors must firstly descript the difference of heat stress in the HHZ and IR36 cultivar (lines 84-88), then start to why you need to carry out this study (lines 82-84).

- METHODS section, “high- temperature” need to remove one space (line 111). “Olympus DP70 digital camera system” (line 126) need to provide the information of company such as (Olympus Corp, Tokyo, Japan). For cDNA library construction by using TruSeq RNA Sample Preparation Kit, authors only followed the manufacturer's instructions, so authors don’t need to describe in detail the protocol (lines 134-146). The line space from 140 to 142 is larger than others. Need to provide the information resources of GO and KEGG enrichment analysis in line 162 and line 167. In addition, Line 109-110, 164-166 contain much information which must be moved to RESULTS section.

- RESULTS section, “Sequencing statistics” change to “Transcriptome analysis.” (line 194). “Go terms” change to “GO terms” (line 242). “Seventeen” change to “17” (line 250).

- DISCUSSION section, “GO enrichment analysis revealed that the DEGs were commonly enriched in response to heat, stress and temperature stimuli in the biological process category” descripted in Fig.3? “heat-tolerant” change to “heat-resistant” (line 336). “hsf” change to “HSF” (line 340). All “et al.” change to “et al.” (italicize) such as in line 343 and 348.

- REFERENCE section, all the references need to list full title of the Journal such as 438, 441, 444, 447…

- FIGURES, In Fig. 3, the position of texts (HHZ_32_vs_HHZ_40 and IR36_32_vs_HHZ_32) need to be modify. In Fig. 9 (a), two DEGs have same name (BGIOSGA015767).

Reviewer 3 ·

Basic reporting

Scientific quality and integrity of the manuscript is satisfactory.

Experimental design

Although the methodology opted to testify the said objectives is well equipped, but still some explanations need to be added, especially the heat stress treatments and the natural environmental variations. The statistical methods and data analyzing tools need to be mentioned (see attached files).

Validity of the findings

The results are valid as per mentioned methodology.

Additional comments

For detailed comments, please refer to the attached comments file.

Annotated reviews are not available for download in order to protect the identity of reviewers who chose to remain anonymous.

---

## Round 0.2 · Minor Revisions

Dear Authors,
Three reviewers and I agree that you made significant improvement in your revised manuscript. But there are still grammar issue or items you did not express clearly. I am attaching my edits in the attached PDF file. Please check your manuscript thoroughly for your next revision. Besides your language issues, please also address the following comments:
1. In Abstract, please also add your main findings from your KEGG analysis.
2. Please provide reference(s) for your statement in line 46-47.
3. In the paragraph starting from line 55, some reference is about other crops and you did not mention it, please be clear about this.
4. Line 109, please provide the ratio for the growing mixture.
5. Line 113, please add the temperature ranges that you mentioned in your response letter.
6. Line 120, please provide information for your growth chamber.
7. Line 147, please provide details on how you measured glumes, including what part of the head, how many from each head.
8. Line 201-211, you only talked about reduction rates, but some are not significant. Please include those information. If you mention that it is significant, you need to add P value.
9. Line 213, you mentioned "greater reduction", did you analyze it statistically? Is it significantly different?
10. Line 228, it should be Table 2, not S4. Table 4S was not mentioned your manuscript. You need to talk about it somewhere. What does Q20, Q30 mean? You need to put notes for these terms. Also there is no Q20(bp), please add it for consistency. You listed both Q20 and Q30, what is your quality standard? Please be clear on that.
11. Line 290, since you have listed all the numbers, figure 8 might not needed.
12. Line 293, where is "HAP" from? I did not see it in the descriptions of 11 TF families.
13. Line 334-335, numbers didn't add up. In the Table S1, you have 28, but here it seems like that you have 30.
14. Please provide references for "Previous studies". What stages did they conduct their studies when you claimed that these genes might not respond in young panicles?
15. Line 367, what "DEGs" are you talking about here? Please also discuss why HSP has no difference in your study?
16. Line 373, "species" is not an appropriate word here.
17. Line 410, why did five of them upregulated, one downregulated? It seems like you mostly discuss the ones that agrees with your findings, but those one that did not agree with yours. Like in line 436, you mentioned upregulated ones, how about those downregulated ones?
18. Line 444, please explain why this is downregulated.
19. Line 470-480, did upregulation of these genes promote heat tolerance or what? The whole paragraph is not clear and needs more clarification.
20. Line 478, this paragraph is talking the same thing as the last paragraph, they should not be split into two paragraphs.
21. Line 483, You mentioned "OsSUS3" in previous studies. But how did it relate to you study? Please discuss it.
22. Line 489, Please discuss that how you can carry out MAS with so many genes you found in your study?
23. You mentioned BGIOSGA031385 at line 330, it seems special. But you did not discuss it later. Please add discussion for this gene.
24. All the table titles should be aligned from the left.
25. Please provide figure captions. I can't find them anywhere.
26. It seems like you have a lot figures and tables. My suggestion is to put pictures from Figure 1, 2, and 3 together as Figure 1. You don't need those graphs in Fig 1, 2, 3. Instead you can put them in numbers as you did in Table S2. Table S2 is important, you should put it as Table in the context. Table 1 in your manuscript is not that important and you can put it as supplemental table. So, all the numbers can be combined and put it in as Table 1 in your revision.

[]

Reviewer 1 ·

Basic reporting

I noticed the improvement in the revision.

Experimental design

looks good in the revision.

Validity of the findings

conclusions seems fine though novelty not improved much.

Additional comments

Thanks for addressing all my comments in revision, and I think the manuscript is in good shape for accepting.

Reviewer 2 ·

Basic reporting

no comment

Experimental design

no comment

Validity of the findings

no comment

Additional comments

Comments to the Author
The revisions made in the current manuscript appropriately address all of my prior comments and it is more clear and concrete, and I am pleased to now recommend acceptance.

Minor comments:
Some sections are still not uniform in the manuscript such as line 66, 114-118, 153-157, I suggest you to go through the manuscript format again carefully and do changes following the guidelines.

Reviewer 3 ·

Basic reporting

Satisfactory.

Experimental design

Satisfactory.

Validity of the findings

Satisfactory.

Additional comments

Authors have revised well the manuscript and addressed the comments. However, authors have tried sometimes just to copy and paste the response to reviewer for certain comments from one reviewer response to other. Authors must be careful in typo errors in revised draft and it is recommended that authors must again need to get proofread their revised draft from the English experts, as it seems that revised draft has not been checked by native speaker. As authors have claimed in the response letter that revised changes are highlighted with red font color, however this was done nowhere. Be careful!

---

## Round 0.3 · Minor Revisions

Please check the whole manuscript to be sure that it is consistent to have space or not before and after those signs such as "<".
Line 52 (in tracked version), delete space before ";" and add space after ";"
Line 114, 381, 486, delete space before "( )"
Line 237, delete space before "on"
Line 332, add space before "including"
Line 498, delete "to"
Line 561-564, delete "differential expression of", "study of', "study of"
Line 758, Replace "Figures" with "Figure legends"
In the Table 1 note, what is NT and HT? you should add temperature in parenthesis to indicate what they are since you only have temperature number in the table.

Before submission, please double check whole manuscript carefully and correct any grammar or editorial errors that you might catch.

---

## Round 0.4 · Minor Revisions

The manuscript has been greatly improved. However, some statements are overstated and it sounds like authors have found proven mechanism for heat tolerance. This is not true. Therefore, please modify the statements at lines 36-39, 519-521, 522-525. For lines 250-252, your statement might be right, but this is not for certain since you are not using two isogenic lines for comparison. Please change the tone for this sentence too.

---

## Round 0.5 · Minor Revisions

A few more editing might be needed as following:
Line 41: change "was associated with" to "might have".
Line 253: add "might have" before "played".
Line 254: change "were" to "might be".
Line 531: change "was associated with the inhibition of" to "might have inhibited the".

---

## Round 0.6 · accepted · Accept

One minor revision is needed: remove "of" in line 514.

Please take the opportunity in production to read through your whole manuscript carefully and correct any grammar or editorial errors that you might catch.